# Aptamer Affinity-Bead Mediated Capture and Displacement of Gram-Negative Bacteria Using Acoustophoresis

**DOI:** 10.3390/mi10110770

**Published:** 2019-11-11

**Authors:** SangWook Lee, Byung Woo Kim, Hye-Su Shin, Anna Go, Min-Ho Lee, Dong-Ki Lee, Soyoun Kim, Ok Chan Jeong

**Affiliations:** 1PCL Incorporated, Seoul 08510, Korea; skim@pclchip.com; 2Institute of Digital Anti-Aging Health Care, Inje University, Gimhea 50834, Korea; ioio109@naver.com; 3Department of Chemistry, Sungyunkwan University, Suwon 16419, Korea; hyesu_1221@nate.com (H.-S.S.); dklee0318@gmail.com (D.-K.L.); 4School of Integrative Engineering, Chung-Ang University, Seoul 06974, Korea; anna@itsannastyle.com (A.G.); mhlee7@cau.ac.kr (M.-H.L.); 5Department of Biomedical Engineering, Dongguk University, Seoul 10326, Korea; 6Department of Biomedical Engineering, Inje University, Gimhea 50834, Korea

**Keywords:** aptamer, acoustophoresis, microfluidics, gram-negative bacteria

## Abstract

Here, we report a simple and effective method for capturing and displacement of gram-negative bacteria using aptamer-modified microbeads and acoustophoresis. As acoustophoresis allows for simultaneous washing and size-dependent separation in continuous flow mode, we efficiently obtained gram-negative bacteria that showed high affinity without any additional washing steps. The proposed device has a simple and efficient channel design, utilizing a long, square-shaped microchannel that shows excellent separation performance in terms of the purity, recovery, and concentration factor. Microbeads (10 µm) coated with the GN6 aptamer can specifically bind gram-negative bacteria. After incubation of bacteria culture sample with aptamer affinity bead, gram-negative bacteria-bound microbeads, and other unbound/contaminants can be separated by size with high purity and recovery. The device demonstrated excellent separation performance, with high recovery (up to 98%), high purity (up to 99%), and a high-volume rate (500 µL/min). The acoustophoretic separation performances were conducted using 5 Gram-negative bacteria and 5 Gram-positive bacteria. Thanks to GN6 aptamer’s binding affinity, aptamer affinity bead also showed binding affinity to multiple strains of gram-negative bacteria, but not to gram-positive bacteria. GN6 coated bead can capture Gram-negative bacteria but not Gram-positive bacteria. This study may present a different perspective in the field of early diagnosis in bacterial infectious diseases. In addition to detecting living bacteria or bacteria-derived biomarkers, this protocol can be extended to monitoring the contamination of water resources and may aid quick responses to bioterrorism and pathogenic bacterial infections.

## 1. Introduction

Bacterial infections remain a major public health and food safety concern [1]. Gram-negative bacteria cause infections, including pneumonia, bloodstream infections, wound and surgical site infections, and meningitis in healthcare settings [2]. Many gram-negative bacteria are resistant to multiple drugs and are becoming increasingly resistant to most available antibiotics [3,4,5]. These bacteria can develop new mechanisms of drug resistance and pass the genetic material underlying drug resistance to other bacteria [6]. These microorganisms, therefore, have great clinical importance in hospitals because they put patients in the intensive care unit (ICU) at high risk and lead to high morbidity and mortality [7].

Rapid, selective, and sensitive technologies for the detection of pathogenic bacteria are necessary for clinical diagnosis, disease control, environmental monitoring, and food safety. Both the detection and identification of bacteria continue to rely on conventional methods, including culture and colony counting of bacteria, immunology-based methods (e.g., ELISA) based on antibody-antigen interactions, and polymerase chain reaction (PCR) involving amplification of DNA [8]. Although these approaches are powerful and error-proof, they are typically laborious, complex, and time-consuming. Moreover, regarding microbiological culture and isolation of the pathogen, it takes five to seven days to confirm the results for some pathogenic organisms [9].

Various microfluidic platforms have been developed to separate bacteria from clinical and environmental samples, including dielectrophoresis, magnetophoresis, bead-based extraction, filtering, centrifugal microfluidics, and methods based on inertial effects and surface acoustic waves (for a comprehensive review, see Wu et al. [10]). Generally, these methods are limited in terms of sample volume, throughput, recovery of bacteria, the efficiency of cell or lysis debris removal, or the need for specific affinity for pathogen capture [11]. To improve binding specificity, affinity bead-based cell sorting methods, which utilized affinity micro-bead for capturing and extraction of bacteria have been developed [12]. The affinity bead-based method generally utilized several physical forces such as acoustic, magnetic, and dielectrophoretic forces to separate the bead bund target from the sample efficiently [13]. This method can be a powerful tool to separate small particle (<1 µm), such as peptides, viruses, and bacteria since the small size of particles are less affected by the applied physical force for manipulating the particles, and can be extracted using affinity microbeads that bind selectively to the particle of interest [13,14].

Most affinity bead-based methods use antibodies as an affinity probe. Although over the decade antibodies made profound contributions to a wide range of fields owing to their diagnostic and therapeutic applications, aptamers have high potential as diagnostic and therapeutic tools, with many advantages when compared with antibodies, including their smaller size, feasibility to access to biological environments, excellent stability, low cost, and higher reproducibility of nucleotide production. These valuable properties make aptamers flexible and powerful tools for diagnostic and therapeutic purposes [15,16].

The term aptamer refers to a short, single-stranded nucleic acid molecule that folds into a three-dimensional structure for efficient binding to a specific target with high affinity [17,18]. Aptamers can recognize target epitopes with high selectivity and specificity via the screening technique of systemic evolution of ligands by exponential enrichment (SELEX) [18,19]. Following the establishment of the SELEX procedure in 1990 [19], many aptamers have been generated against a variety of targets, from small chemical compounds to large multidomain proteins [20,21], as well as whole cells such as bacteria and cancer cells [22,23]. Especially, the cell-SELEX process, which is used to select probes capable of recognizing molecular signatures on the surface of diseased cells, has attracted a great deal of interest for biomarker discovery, as well as cancer diagnosis and therapy [20,21,22,23].

Microfluidic acoustophoresis systems have attracted a great deal of attention for flow through, label-free, non-contact cell separation with reasonable throughput [24,25,26]. Acoustophoresis refers to the migration of particles via ultrasonic sound waves. When a sound wave drives into a microchannel, it generates acoustic radiation forces that act on suspended objects in the nodal (or antinodal) plane according to their intrinsic properties, such as size, density, and compressibility [27]. In an acoustophoretic channel with sheath flow, the particles are first laminated to the sides of the channel via the central sheath flow of a clean buffer and then merged into the clean buffer due to the acoustic radiation force, based on their acoustophoretic mobility. This label-free cell separation method can operate independently of the biochemical and electrical properties of the suspending medium [26]. Therefore, this method has been used extensively to separate, concentrate, or wash various biological samples, including plasma, human blood, cell culture medium, and raw milk [17,18,19]. 

Recently, the acoustic properties of functionalized affinity microbeads have been exploited for acoustophoretic separation of small particles (<1 µm), such as peptides or viruses, which are less affected by the primary acoustic radiation force and can be extracted using affinity microbeads that bind selectively to the particle of interest [27]. Augustsson et al. reported a microfluidic device to separate phages using antigen (grass pollen allergen; Phl p5)-coated microbeads from commonly used phage display libraries. Park et al. developed an acoustofluidic device as a new screening method using a prostate-specific antigen (PSA)-coated microbeads for selection of aptamer based on an acoustofluidic separation (acoustophoresis) technique [28]. In addition, Shields et al. demonstrated a new acoustofluidic sorting method using elastomeric affinity particles in order to capture and transport cells into antinode (general channel wall) of an acoustic standing wave [29].

Here, we report the development of a microfluidic acoustophoresis protocol that enables efficient and rapid separation of gram-negative bacteria from culture media using aptamer-modified microbeads. Our system enhances the separation efficiency via the incorporation of a long, square-shaped microchannel. Based on this architecture, the simultaneous excitation of two orthogonal resonances can generate a two-dimensional (2-D) acoustic standing wave via single piezoelectric actuation, tightly focusing particles, and cells on nodal (antinodal) points. Two trifurcated channels located at both the inlet and outlet regions make it possible to perform separation, purification, and concentration procedures simultaneously. 

We utilized the GN6 aptamer [30], which we previously selected using cell-SELEX, to achieve high specific binding affinity to various species of gram-negative bacteria, but not to gram-positive bacteria. Biotin-tagged GN6 aptamer was immobilized on streptavidin-coated polystyrene (PS) microbeads. The GN6-immobilized microbeads were initially incubated with cultured bacterial solutions and then injected into the acoustophoretic channel for simultaneous separation and washing. As the sample mixture entered the acoustic standing wave field, the microbeads bound to target bacteria migrated across the central buffer interface and exited the system through the central outlet, whereas the unbound bacteria remained in the original buffer stream along the sidewalls and were removed through the side outlets, as shown in Figure 1. The migration of beads via the acoustic wave force field, from the sidewall to the center of the clean buffer, enabled simultaneous washing and separation in the continuous flow. The efficacy of the device was characterized using three parameters, i.e., the recovery ratio, purity, and concentration ratio (Figure 2). To form a laminar flow in the microchannel, the sample and buffer were infused into each inlet using syringe pumps with a total flow rate of 500 µL/min (sample flow: 150; buffer flow: 350). The separated target bacteria and non-targets were collected in the target outlet and waste outlet, respectively, via flow control. The recovery rate and purity were >90%, with a concentration factor of 25. 

As proof of principle, we tested the separation performance of the microfluidic acoustophoresis device using five different gram-negative bacteria and five different gram-positive bacteria. Streptavidin-coated microbeads were initially incubated with biotin-labeled gram-negative bacteria-bound DNA aptamer (GN6) for 30 min in room temperature. The incubated mixture was then injected into the acoustophoretic channel for simultaneous separation and washing. As the sample mixture entered the acoustic standing wave field, the microparticles with target-bound DNA fragments migrated across the central buffer interface and exited the system through the central outlet, whereas the unbound bacteria and other contaminants remained in the original buffer stream along the sidewalls and were removed through the side outlets. The migration of beads via the acoustic wave force field, from the sidewall to the center of the clean buffer, enabled simultaneous washing and separation in the continuous flow. After the separation of target-bound beads using the device, we examined bound gram-negative bacteria on each microbead in a bead-by-bead manner. This protocol showed high specificity toward all gram-negative bacteria, as well as binding affinity to multiple gram-negative bacteria (but not to gram-positive bacteria). This study may present a different perspective in the field of early diagnosis in bacterial infectious diseases. In addition to detecting living bacteria or bacteria-derived biomarkers, this protocol can be extended to monitoring the contamination of water resources and may aid quick responses to bioterrorism and pathogenic bacterial infections.

## 2. Materials and Methods

### 2.1. Microfluidic Acoustophoresis Chip Design

The acoustofluidic device includes a sample inlet, buffer outlet port, main channel, and outlet channel with a collection outlet port and a waste outlet port (Figure 3). The main channel was fabricated in a layered (glass-silicon-glass) format (Figure 3a) with a cross-section of 0.2 × 0.2 mm. An initial borosilicate glass bottom layer (20 mm × 80 mm × 0.5 mm) was bonded to the middle silicon layer (20 mm × 80 mm × 0.5 mm) by anodic bonding. Using a chemical mechanical polishing (CMP) process, 300 µm of the silicon layer was removed so that the size of the middle silicon layer changed to 0.2 mm. This was followed by silicon patterning using a photoresist (SU-8 3010; Nippon Kayaku, Tokyo, Japan) and deep reactive ion etching to form a microchannel with a cross-section of 0.2 × 0.2 mm. The top layer of borosilicate glass (20 mm × 80 mm × 0.5 mm) was sandblasted to form holes as inlets and outlets, and this layer was bonded to the silicon layer through anodic bonding. The 20 × 40 mm piezoelectric transducer (C-213; Fuji Ceramics, Shizuoka, Japan) was glued to the borosilicate glass layer along the microfluidic channel using cyanoacrylate glue (AD100; 3M). The sample inlet and buffer inlet channels had widths of 0.4 and 0.1 mm, respectively. The collection outlet and waste outlet channels had widths of 0.2 and 0.4 mm, respectively. The real picture of the device is displayed in Figure 3b.

### 2.2. Acoustophoresis Setup

The piezoelectric actuator was driven by a single-channel functional generator (AFG-2225; GW Instek, Seoul, Korea) and the generated signals were amplified using a power amplifier (75A250A; Amplifier Research, Souderton PA, USA) (Figure 3c). The resonance frequency of the actuator was 3.66 MHz, matching the size of the channel (Figure 1). The transducer actuated simultaneous resonance in the chip due to the channel’s square shape, thus focusing cells or particles in the center of the microfluidic channel. The flow rates were controlled by syringe pumps (70-4505 Elite Pump; Harvard Apparatus, MA, USA) mounted with syringes (SS05-LZ; Terumo, Tokyo, Japan) connected to the inlets and outlets of the channel. The inlets and outlets were directly linked to the syringe pumps by polyetheretherketone (PEEK) tubes. To avoid bubble entrapment, the channels were filled with deionized water before initiating the experiment. After device preparation, syringes containing the samples were loaded. 

The separation and concentration processes in the acoustofluidic chip were observed in the device using a microscope (IX-81; Olympus Corp., Tokyo, Japan) and a high-speed camera with a frame rate of 1200 fps. These observations were quantified and analyzed based on direct images taken with the high-speed camera (FASTCAM Mini; Photron, Tokyo, Japan).

### 2.3. Bacterial Strains and Culture

Five gram-negative and five gram-positive bacteria were chosen for the experiments, as shown in Table 1. All bacteria were purchased from the Korean Collection for Type Cultures (KCTC). *Escherichia coli* DH5α, *E. coli* (KCTC2571), *Staphylococcus epidermidis* and *Staphylococcus pasteuri* were cultivated at 37 °C in Luria–Bertani (LB) medium, *Enterobacter cloacae* and *Bacillus megaterium* (KCTC 1021) were grown at 30 °C in nutrient broth (NB) medium, *Pseudomonas pictorum* and *Sphingomonas insulae* were grown at 25 °C in LB medium, *Enterococcus thailandicus* was grown at 37 °C in Lactobacillus Man, Rogosa, and Sharpe (MRS) broth, and *Listeria grayi* was grown at 37 °C in brain heart infusion (BHI) medium. All of these bacteria were cultured under aerobic conditions up to an optical density at 600 nm (OD_600_) of 0.4, followed by centrifugation (~14,200 g) for 10 min at 4 °C, and washed twice with Tris-HCl buffer (50 mM Tris, pH 7.4, 1 mM MgCl_2_, 5 mM KCl, 100 mM NaCl). The washed bacteria were resuspended in binding buffer (50 mM Tris, pH 7.4, 5 mM MgCl_2_, 5 mM KCl, 100 mM NaCl, 1 mg/mL bovine serum albumin [BSA], 0.1 mg/mL salmon sperm DNA, 0.1 mg/mL yeast tRNA).

### 2.4. Microbeads and Immobilization of Aptamer onto Microbeads

10 µm, 3 µm and 1 µm diameter of polystyrene microbeads, and 10 µm streptavidin-coated microbeads were purchased from *Bangs Laboratiries, Inc.* (PS07001, PS05002, PS04001, and CP01007, respectively). Streptavidin-coated microbeads (10 µm, CP01007; Bang Laboratories, Inc., Fishers, IN, USA) were resuspended in vials (or vortex-mixed for 20 s) and 250 µL aliquots were transferred into 1.5-mL tubes. Then, 250 µL of biotinylated DNA aptamer was added to the tubes, making a final concentration of 50 pmol; the mixture was incubated for 30 min at room temperature, followed by centrifugation (10,000 rpm) and washing twice with Tris-HCl buffer. For blocking, 10 µL of BSA (100 mg/mL) and 5 µL of yeast tRNA (10 mg/mL) were added to the tubes, followed by incubation for 30 min at room temperature. Finally, the aptamer-modified microbeads were washed twice by centrifugation (10,000 rpm) in Tris-HCI buffer. 

### 2.5. Acoustophoresis

Acoustophoresis refers to the manipulation of suspended particles in a fluid by acoustic radiation forces in a continuous flow microchannel. This manipulation can enrich particles, transfer them from one carrier fluid to another, or distinguish them according to their size, density, or compressibility [31]. The acoustic radiation forces are produced by vibrating the microfluidic device using a piezoelectric actuator, and these forces create resonance patterns within the fluid. The particles in suspension experience a force in the direction of the pressure gradient formed by the resonance pattering, transferring them to either pressure minima or maxima depending on the acoustic properties. In an acoustophoresis system, larger, denser, and less compressible cells move faster into the nodal (or antinodal) plane of the standing wave according to Equations (1) and (2):*F*_rad_ = 4π*a*^3^ϕ*k_y_*E_ac_sin(2*k_y_y*)(1)
where,
ϕ = (κ_o_ − κ_p_)/3κ_o_ + (ρ_p_ − ρ_o_)/(2ρ_p_ + ρ_o_)(2)

*F*_rad_ is the acoustic radiation force acting on the particle, *a* is the radius of the cell, ϕ is the acoustic contrast factor, *k_y_* is the wave number, E_ac_ is the acoustic energy density, *y* is the distance from the wall along the axis of the standing waves, κ_o_ and κ_p_ are the isothermal compressibility of the fluid and particles, respectively, and ρ_p_ and ρ_o_ are the densities of the particles and fluid, respectively.

## 3. Results and Discussion

The main parameters affecting the magnitude of the acoustic radiation force are the radius of the particle, *a*, and the acoustic energy density, E_ac_. The size sensitivity of the acoustic radiation force made it possible to fractionate particles by size. The trifurcation of the channel laminated the particles toward the channel sidewalls. Under conditions of acoustic manipulation, larger flowing particles move faster to central pressure nodes than smaller flowing particles. 

By infusion of a cell-free buffer liquid through the buffer inlet, the infused bacteria and counteraffinity bead mixture were hydrodynamically laminated close to the sidewalls. A single node standing wave deflected the cells toward the channel center, such that the final position at the end of the channel was determined by the acoustic contrast and size of the cells. To avoid medium switching, in which the liquid containing the sample and the cell-free liquid exchange locations, the cell-free liquid must have equal or greater acoustic impedance than the sample-containing liquid. In the experiments reported here, the solution used for suspending the particles was the same as the cell-free central sheath liquid. The 2-D focusing ensured that all of the particles were located in the same flow velocity regime within the parabolic flow profile, making the separation more efficient. The particles all had the same retention time in the separation zone.

To investigate the separation performance of the device, we used a mixture of PS particles of 10 µm, 3 µm, and 1 µm in diameter, corresponding to the sizes of aptamer-modified beads (10 µm) and bacteria (3 µm and 1 µm), respectively. The PS sample mixture with a concentration of ~2 × 10^5^ particles/mL (10µm beads) and ~4 × 10^6^ particles/mL (3 µm & 1 µm beads) were infused into the sample inlet port. PS particles of 10 µm were collected at the target outlet; the reminder (1 and 3 µm) were discarded at the waste outlet (Figure 1). As shown in Figure 3, the mixed sample was transferred into a 10-mL syringe (SS05-LZ; Terumo) and directly linked to the sample inlet port. Similarly, a syringe containing a cell-free buffer was linked to the buffer inlet port. The syringe pumps were infused with the sample and cell-free buffer at flow rates of 200 and 300 µL/min, respectively, for a total infused flow of 500 µL/min. The rates of withdrawal from the waste outlet port were set to 400, 450, and 475 µL/min, for flow rates at the collection outlet of 100, 50, and 25 µL/min, resulting in volumetric concentrations of 5×, 10×, and 20×, respectively. The separated and concentrated target beads (10 µm) were collected in 50-mL centrifuge tubes (352070; Falcon) from the collection outlet (Figure 3). By controlling the flow rate of the syringe pump linked to the waste outlet, target beads could be separated from the mixed sample and concentrated. As target beads migrate from the sheath layer to the cell-free buffer stream, the cells could be purified.

As shown in Figure 2, a mixture of 10- and 1-µm microbeads and bacteria (*E. coli* KCTC2571) was injected into the acoustofluidic chip, and the separation performance of the chip was evaluated under three different flow conditions. The measured recovery, purity, and concentration factor are shown as a function of the proportion of the number of particles collected at the target outlet compared to the total number of particles collected (at the target outlet and waste outlet; Figure 4) The recovery values were 98% ± 2.1%, 98% ± 2.3%, and 90% ± 9.2%, while the purity values were 97.1% ± 3.1%, 97.8% ± 2.1%, and 99.5% ± 0.4%, respectively. The concentration factors were 5 ± 2.8, 10 ± 1.9, and 20.9 ± 3.9, for volumetric concentrations of 5×, 10×, and 20×, respectively.

The device showed recovery >98% up to 10 × volumetric concentration. However, the recovery decreased to 90% at a volumetric concentration of 20×, because the strong withdrawing force from the waste outlet interrupted the acoustic radiation force, causing some of the 10-µm beads to flow into the side channel. The performance tests of the chip were conducted using a direct observation of the flowing particles in the acoustofluidic chip. Although recovery is up to 98% in the chip, recovery rate (yield) may be worse due to several factors such as sedimentation of bead in the syringe or clogging them into connection tube from syringe to the chip. 

Based on the size separation performance, we tested this microfluidic acoustophoresis to capture and displace the gram-negative bacteria from culture samples. In this experiment, we fixed the condition of the fluidic operation of the chip at a waste outlet flow rate of 450 µL/min (10× volumetric concentration), because this showed the best fluidic performance with regard to purity and recovery. We first tested the capturing and displacement performance of the chip using five different gram-negative bacterial solutions (Table 1). Each bacterium was mixed with aptamer-modified microbeads (10 µm) and incubated for 30 min at room temperature. The mixture was injected into the sample inlet port of the chip for acoustophoretic separation. As the bacteria-bound microbeads and unbound bacteria were separated mainly based on their size difference, some of the unbound microbeads could flow into the sample collection port. As a negative control, we performed the separation experiment using a mixture of beads and five different gram-positive bacteria (Table 1). 

After the fluidic operation, the target sample and waste were collected into tubes and the waste outlet syringe, respectively. Samples of 10 µL were taken from the sample collection tubes and dropped onto slide glasses for observation of fluorescence intensity using a fluorescence microscope. The Figure 5a displayed the images of bacteria bound on the beads. A number of gram-negative bacteria (ex. E-coli DH5α was bound on each bead while no or less (one or two) Gram-positive (ex. Listeria grayi) bacteria were bound on the beads. The fluorescence intensity of all bacteria on the beads (>100 beads) was measured using Image J software (NIH; Figure 5). The observation experiments were repeated three times. All five gram-negative bacteria were bound with high affinity. In contrast, significantly reduced binding affinity was observed against all gram-positive bacterial strains tested. Signal intensity was significantly different between gram-negative and gram-positive bacteria (*p* < 0.0001, *t*-test).

## 4. Conclusions

We developed an acoustophoretic microfluidic device for capturing and displacement of gram-negative bacteria from culture samples based on their size and shape in a rapid, label-free, continuous running manner using aptamer-modified microbeads. A long, square-shaped microchannel allowed for 2-D acoustophoresis with a simpler design and higher cost-effectiveness compared to those reported previously. The proposed acoustofluidic device showed excellent recovery, purity, and concentration factor. We summary of comparison of our method to other previously reported ones in Table 2. Further studies will focus on the parallelization of the system for the preparation of large amounts of liquid samples for high-speed processing.

## Figures and Tables

**Figure 1 micromachines-10-00770-f001:**
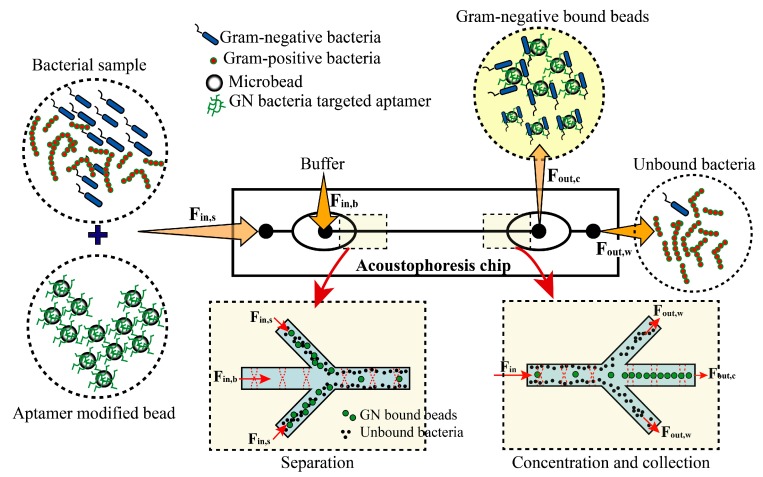
Separation and collection of target microbeads in the microchannel by acoustophoresis. Bacteria were mixed with aptamer-modified microbeads (10 µm) and incubated for 30 min at room temperature. The mixture was injected into the sample inlet port of the chip for acoustophoretic separation. As the separation of the bacteria-bound microbeads and unbound bacteria were mainly based on the difference in their sizes, some of the unbound microbeads flowed into the sample collection port.

**Figure 2 micromachines-10-00770-f002:**
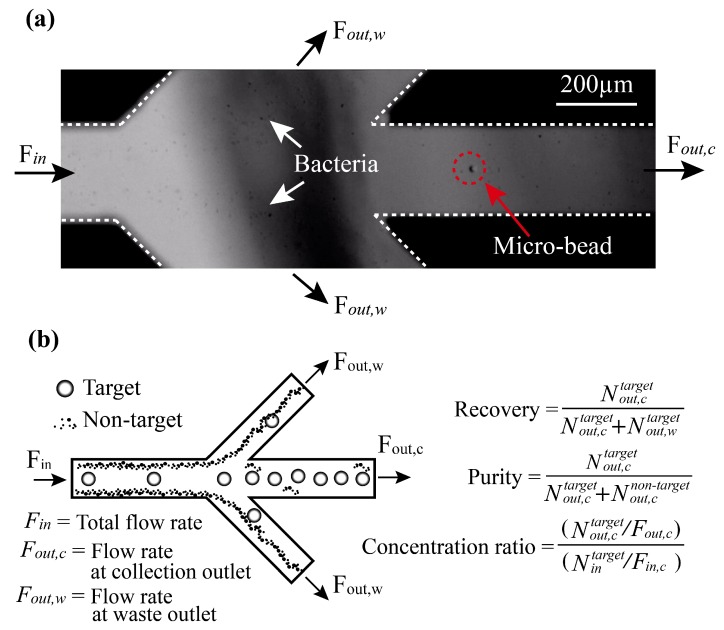
Separation of aptamer-bound beads. (**a**) A high-speed camera (5000 fps) was used to observe the separation of cells. As a proof-of-principle demonstration, a mixture of aptamer-modified beads (9.6 µm) and bacteria (<1 µm) was infused into the sample inlet by one of the syringe pumps at a controlled flow rate of 500 µL/min (flow speed ~0.4 cm/s). The target and non-target were separated by acoustic radiation forces. The target was collected at the collection outlet and the non-target was moved to the waste outlet. (**b**) Definitions of the parameters used for evaluation of separation performance of the acoustofluidic device: *F_in_*, total flow rate; *F_out,c_*, flow rate at the collection outlet; *F_out,w_*, flow rate at the waste outlet; *N_in_*, total number of targets per second; *N_out,c_*, number of targets per second at the collection outlet; $N_{out,c}^{target}$, number of targets per second at the collection outlet; $N_{out,c}^{nontarget}$, number of non-targets per second at the collection outlet; $N_{out,w}^{target}$, number of targets per second at the waste outlet; $N_{out,w}^{non\text{-}target}$, number of non-targets per second at the waste outlet.

**Figure 3 micromachines-10-00770-f003:**
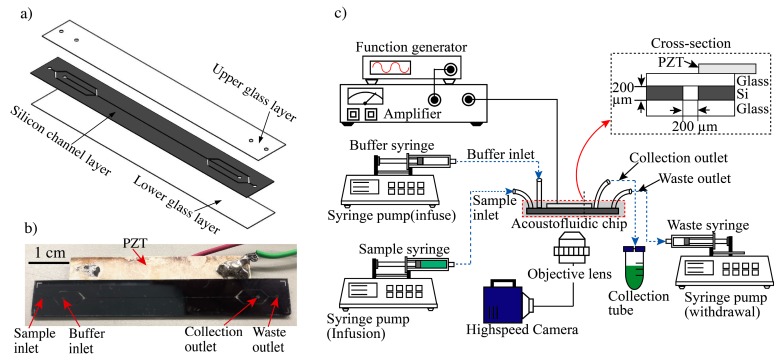
Experimental setup of the acoustofluidic system for cell separation and collection. (**a**) Structure of acoustophoresis microdevice. (**b**) Real picture of the device. (**c**) A sample syringe pump and buffer syringe pump infuse sample or buffer solution into the inlet ports. A separate syringe pump withdraws fluid designated as waste. Separated target samples are collected through the collection tube for downstream analysis. A function generator with a power amplifier applies the AC power to the piezoelectric actuator to generate an acoustic radiation force in the microchannel. This force allows for particle focusing and separation of samples. A high-speed camera module allows for visualization of the separation.

**Figure 4 micromachines-10-00770-f004:**
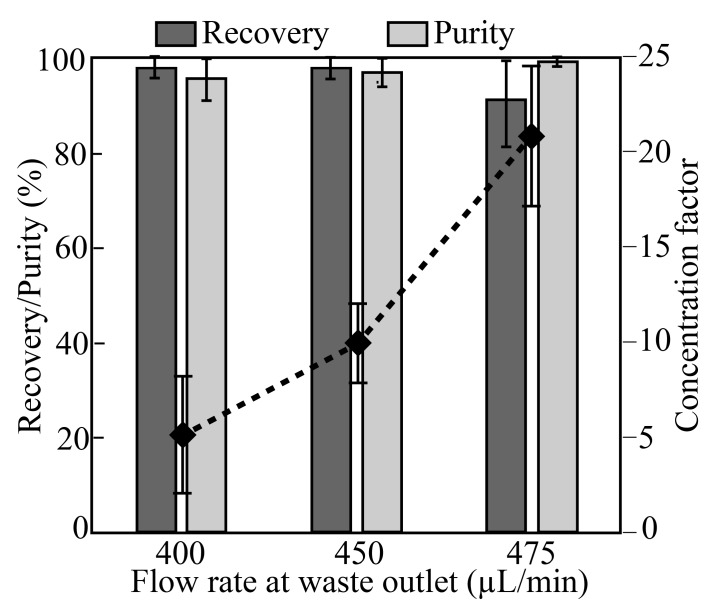
Separation performance of the device. Separation of target microbeads (10 µm) from a mixture of microbeads (1 and 3 µm) and bacteria (*E. coli*). Separation performance was measured based on the recovery and purity parameters defined in Figure 3b. The total infusion flow rate of the inlets (sample inlet and buffer inlet) was 500 µL/min (~0.4 cm/s flow speed), while the syringe pumps of the waste outlet had withdrawal rates of 400 µL/min (5× volumetric concentration), 450 µL/min (10× volumetric concentration), and 475 µL/min (20× volumetric concentration). The recovery rate and purity were determined by repeated experiments (*n* = 3).

**Figure 5 micromachines-10-00770-f005:**
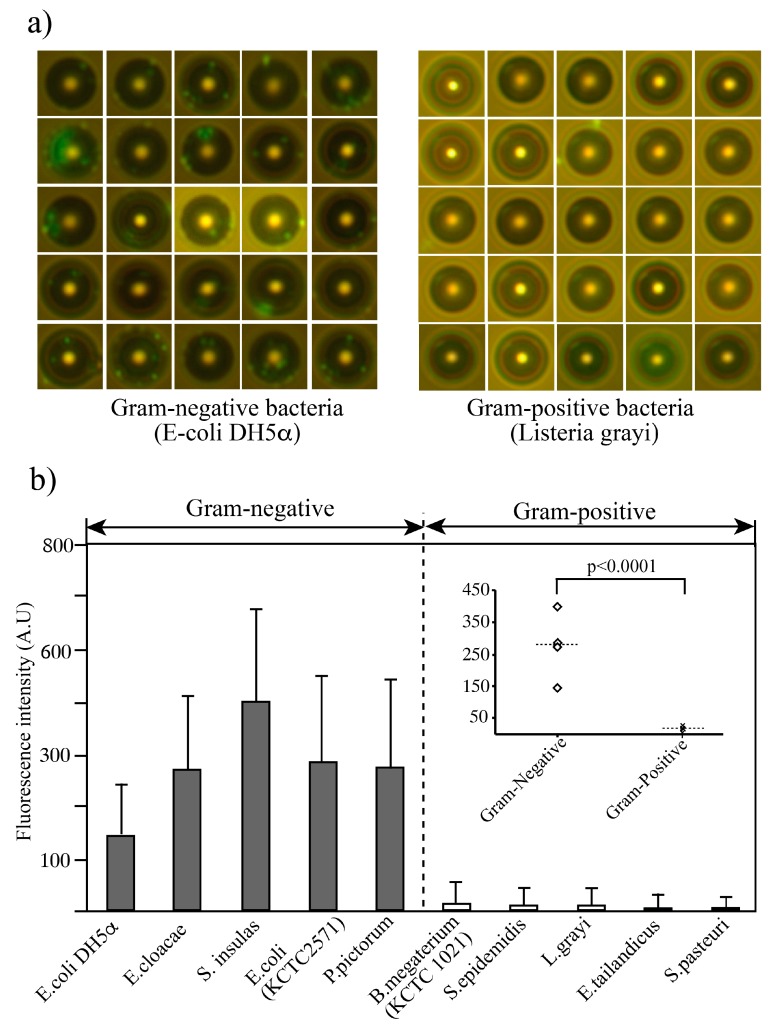
Binding profile of aptamer-modified microbeads against gram-negative and gram-positive bacteria. (**a**) Collected beads from collection outlet. Gram-negative bacteria are bound on the aptamer-coated bead while Gram-positive bacteria shows no or less binding on the beads. Fluorescent intensities were measured each bead and summarized the intensities. After fluidic operation, the target sample and waste were collected into tubes and the waste outlet syringe, respectively. Then, 10-µL samples were taken from the sample collection tubes and dropped onto slide glasses for observation of fluorescence intensity using a fluorescence microscope. (**b**) The fluorescence intensity of the beads (>100 beads) was measured using Image J software (NIH). Data are shown as means ± SD of three independent experiments. Signal intensity was significantly different between gram-negative and gram-positive bacteria (*p* < 0.0001, *t*-test).

**Table 1 micromachines-10-00770-t001:** Gram-negative and gram-positive bacteria.

Gram-Negative Bacteria	Gram-Positive Bacteria
*Escherichia coli* DH5α	*Bacillus megaterium* (KCTC 1021)
*Enterobacter cloacae*	*Staphylococcus epidermidis*
*Sphingomonas insulae*	*Listeria grayi*
*Escherichia coli* KCTC 2571	*Enterococcus thailandicus*
*Pseudomonas pictorum*	*Staphylococcus pasteuri*

**Table 2 micromachines-10-00770-t002:** Bead based bacteria separation methods.

Bead-Labeling	Microfluidic Method	Affinity Legend	Sample Throughput	Purity	Target Cell Recovery	Reference
Label-free	Inertia	N.A	8 mL/min	99.87%	62%	[32]
Membrane filter	N.A	500 µL/min	–	89.9%	[33]
SSAW	N.A	1 µL/min	95.65%	N.A.	[34]
BAW (Impedance matches)	N.A	400 µL/min	90%	99%	[35]
DEP	N.A	0.5 µL/min	–	87.2%	[36]
Affinity bead	DEP & Magnetophoresis	Antibody	8.3 µL/min	95%	98%	[37]
Magnetophoresis	Antibody	150 µL/min	80%	–	[38]
BAW (This work)	Aptamer	500 µL/min	99.5%	98% (Bead recovery)

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
