# Peer review of "Aptamer Affinity-Bead Mediated Capture and Displacement of Gram-Negative Bacteria Using Acoustophoresis"

_micromachines, 2019, doi:10.3390/mi10110770_

Round 1

Reviewer 1 Report

The authors present an experimental study using aptamer-conjugated beads to separate gram-negative bacteria from gram-positive bacteria. The work is clear, of good quality, and of interest to the readership of Micromachines. However, I have concerns with some of the claims made by the authors regarding originality, as many other bead-based microfluidic technologies have been developed to separate cells. I am also confused about the meaning and significance of the data in Figure 5. Thus, I believe the following clarifications or corrections must be made before this work can be accepted:

Authors should add a definition of “concentration factor” to the abstract. Authors claim “the acoustophoresis microfluidic device also showed binding affinity to multiple strains of gram-negative bacteria, but not to gram-positive bacteria”, but binding was performed outside of the chip. The authors claim: “This study presents a new paradigm for early diagnosis of bacterial infectious diseases”. This is an incredible claim with little substantiation. Frankly, many microfluidic systems have been developed to separate bacteria, and it is not clear what advantages this system has over others. Additionally, this work does not provide any strong analytical evaluation beyond quantification. I suggest removing “new paradigm” both in the abstract and the introduction. In the introduction, the authors suggest that PCR and ELISA lack specificity. This assertion is baseless. PCR and ELISA are gold standards for bacterial detection due to their sensitivity and specificity. The authors mention other methods for microfluidic separation, but they failed to compare their technology to other positive-selection techniques. See the article W. Shields IV, C. D. Reyes, G. P. López, Lab Chip, 2015, 15, 1230–1249 and their discussion “Bead Based Sorting”. Below are some bead-based methods that have come before this study and should be mentioned (none are discussed): Electrics U. Kim and H. T. Soh, Lab Chip, 2009, 9, 2313–2318. Acoustics C. W. Shields IV, L. M. Johnson, L. Gao and G. P. López, Langmuir, 2014, 30, 3923–3927 Magnetics S. Miltenyi, W. Muller, W. Weichel and A. Radbruch, Cytometry, 1990, 11, 231–238 (there are many, but this was the first) While the authors describe positive acoustic contrast bead-based sorting assays, they fail to mention negative acoustic contrast particles. Please refer to studies involving those particles and compare/ contrast. What is the clinical basis for separating gram negative and positive bacteria? This motivation was not mentioned in the article. Why should a clinician care about separating two types of bacteria from a patient? With regards to diagnosis, ELISA and PCR can still provide useful diagnostic information with both cell types present. In the recovery equation, N(target, out, w) is used once, but the value is used twice in the schematic. I think the authors mean to indicate the sum of the two N(target, out, w)’s? The sentence “Microbeads modified with a target molecule were initially incubated with gram-negative bacteria-bound DNA aptamer (GN6) and then injected into the acoustophoretic channel for simultaneous separation and washing” skips the critical step where bacteria are incubated with the microbeads. How were the channel dimensions measured? The values seem to suggest that the dimensions were assumed. Correspondingly, how was resonance determined? Based off of the theoretical dimensions? Or were experiments performed to identify the half wavelength harmonic? Please describe the centrifugation in xG. RPM values vary depending on the centrifuge used. To improve visual clarity, it would help to make the second y-axis in Figure 4 blue. Otherwise, this should be specified in the caption. Despite my best attempt, I do not understand Figure 5 at all. The y-axis is fluorescence intensity, but fluorescence is coming from the beads and not the bacteria, correct? How are the different bacteria then identified and plotted if they are all mixed together and captured on the same beads? Also, isn’t the signal intensity a representation of the purity and not of the number of bacteria? Clearly I am missing something; please clarify.

Reviewer 2 Report

Recommendation: Minor revisions needed as noted. I have thoroughly reviewed the manuscript entitled "Aptamer bead-mediated separation of gram-negative bacteria using acoustophoresis". This manuscript describes a microfluidic acoustophoresis protocol that enables efficient and rapid separation of gram-negative bacteria using aptamer modified microbeads. The manuscript is well written, the experiments are clearly presented the reference list is adequate. The manuscript come with a clear and novel idea that can be promising early diagnosis tool of bacterial infectious diseases, as a result I strongly recommend the publication of the manuscript as it is. Additionally, some suggestions were listed as follows for the authors to improving the work. Minor comments: # 1. I suggest that the authors summarize the specific characteristics (e.g., sample volume, throughput, recovery, purity, yield, etc.) of proposed method in the form of a Table by comparing conventional methods. # 2. I suggest that the author would describe the recovery rate (~yield) of total microbeads in the viewpoint of reusability. Because of gravity, I assume that microbeads in the mixture were sedimented in syringe which can be induced clogging of microchannel before infusing the inlet. # 3. Authors should be modified and unified style. Please read carefully the author guideline in this journal.

Round 2

Reviewer 1 Report

I still have two concerns, the first is minor and the second is major: 

Q2) The authors misunderstood my 2nd concern. My point was that binding was performed outside of the chip, but separation was performed within the chip. The way the abstract was initially written suggested that binding and separation occurred within the chip. Also, the new sentence is problematic. To fix this, the authors can keep their original sentence, but I suggest they modify it to: "the APTAMER-MODIFIED BEADS also showed binding affinity to multiple strains of gram-negative bacteria, but not to gram-positive bacteria". 

Q10) Thanks for clarifying Figure 5. Given this description, I have major concerns that the representation of data is misleading. The authors explicitly state in the text "Next, we evaluated the performance of microfluidic acoustophoresis for separation of gram-negative bacteria from culture samples", but in the rebuttal, the authors state, "We do not mix Gram-positive and Gram-negative bacteria samples". This is not honest scientific representation. The authors should modify their description of Figure 5 to indicate that bacteria were not "mixed" or "separated", but studied independently. And instead of describing bacteria that moved to the pressure node as "separated", they must use a different terminology, such as "shunted" or "displaced". Most cell sorting experts will argue that you are not separating anything if there is only one type of bacteria present in each sample. Alternatively, if the authors insist on keeping the terminology of "separated", they must demonstrate actual separation using mixed bacteria samples. 

Round 3

Reviewer 1 Report

The authors have addressed my concerns.

One last point: the title and text could improve by replacing "capturing and displacement" with either "capture and displacement" or "capturing and displacing", as appropriate.